# Environment-Friendly Ascorbic Acid Fuel Cell

**Md. Mahmudul Hasan** 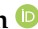

Research Organization for Nano and Life Innovation, Waseda University, Shinjuku, Tokyo 162-0041, Japan; hmahmudulche@gmail.com or hasan@aoni.waseda.jp

**Abstract:** Recently, ascorbic acid (AA) has been studied as an environment-friendly fuel for energy conversion devices. This review article has deliberated an overview of ascorbic acid electrooxidation and diverse ion exchange types of AA-based fuel cells for the first time. Metal and carbon-based catalysts generated remarkable energy from environment-friendly AA fuel. The possibility of using AA in a direct liquid fuel cell (DLFC) without emitting any hazardous pollutants is discussed. AA fuel cells have been reviewed based on carbon nanomaterials, alloys/bimetallic nanoparticles, and precious and nonprecious metal nanoparticles. Finally, the obstacles and opportunities for using AA-based fuel cells in practical applications have also been incorporated.

**Keywords:** ascorbic acid electrooxidation; electrocatalyst; PEM fuel cell; AEM fuel cell; split-pH fuel cell



## 1. Introduction

The current energy system is predominantly fossil-fuel-based and contributes to alarming greenhouse emissions. This resulted in a greater need for alternative energy sources due to rising global energy demands driven by population increase and economic expansion. Modern wind and solar power technologies have developed quickly to provide sustainable and clean energy to meet energy demands. The geographical reliance, irregularity, and high starting costs of these energy generation methods put them at a competitive and economic disadvantage. Battery and fuel cell technologies that use renewable energy are receiving widespread attention in academia and business to overcome fossil fuel dependency [1].

In comparison to other sustainable energy sources, fuel cells are rapidly replacing gas, oil, coal, petroleum, etc. A fuel cell generates electricity from chemical energy and oxygen [2]. Direct liquid fuel cells (DLFCs) provide sustainable, carbon-neutral, and efficient power generation prospects. Transportation, permanent and mobile equipment, as well as a secondary power source are examples of DLFCs application [3]. The energy density of methanol is high, and it is also simple to store and distribute compared to hydrogen. Thus, a direct acidic methanol fuel cell (DMFC) is a preferable alternative for proton exchange membrane fuel cells (PEMFCs), where a cation exchange membrane (CEM) is used. The moderate toxicity of methanol and the catalyst poisoning (CO adsorption) that occurs in acidic DMFCs necessitate the use of expensive anode catalysts [4]. Direct formic acid fuel cells (DFAFCs) are gaining popularity because they have a lower fuel crossover and are less hazardous than methanol [3]. Conversely, it has less energy density and required costly Pt catalysts [3]. Furthermore, DFAFCs are highly vulnerable to CO poisoning on noble metals (Pt and Pd) during long-time operations [5]. Since ethanol is renewable and has a low level of toxicity, direct ethanol fuel cells (DEFCs) have been studied. However, even with precious metals, the ethanol oxidation rate is quite low in acidic environments [6].

Since the invention of anion exchange membranes (AEMs), many fuels can be oxidized efficiently in alkaline conditions compared to an acidic environment. In AEMs, hydroxyl ions are moved from cathode to anode over the membrane with lowered fuel crossover. In addition, the alkaline environment of AEM has benefits for water management, minimal

overpotential requirements, and oxidation and reduction. Significantly reducing the cost of operation is made possible by nonprecious metal (Cu) catalysts in alkaline fuel cells [7]. This ensures the feasibility of DLFCs, as costly noble metals could be replaced. In terms of environmental considerations, the release of $CO_2$ makes commercialization more challenging for alcohol-based DLFC, in both cases of acidic and alkaline. Figure 1 shows the schematic diagram of alcohol-based DLFC fuel cells. An environment-friendly fuel source is highly desirable to generate clean energy. Ascorbic acid (AA) could be used for clean energy generation. This review will provide more information and explore the feasibility of using AA as an alternative environmentally benign fuel for energy conversion.

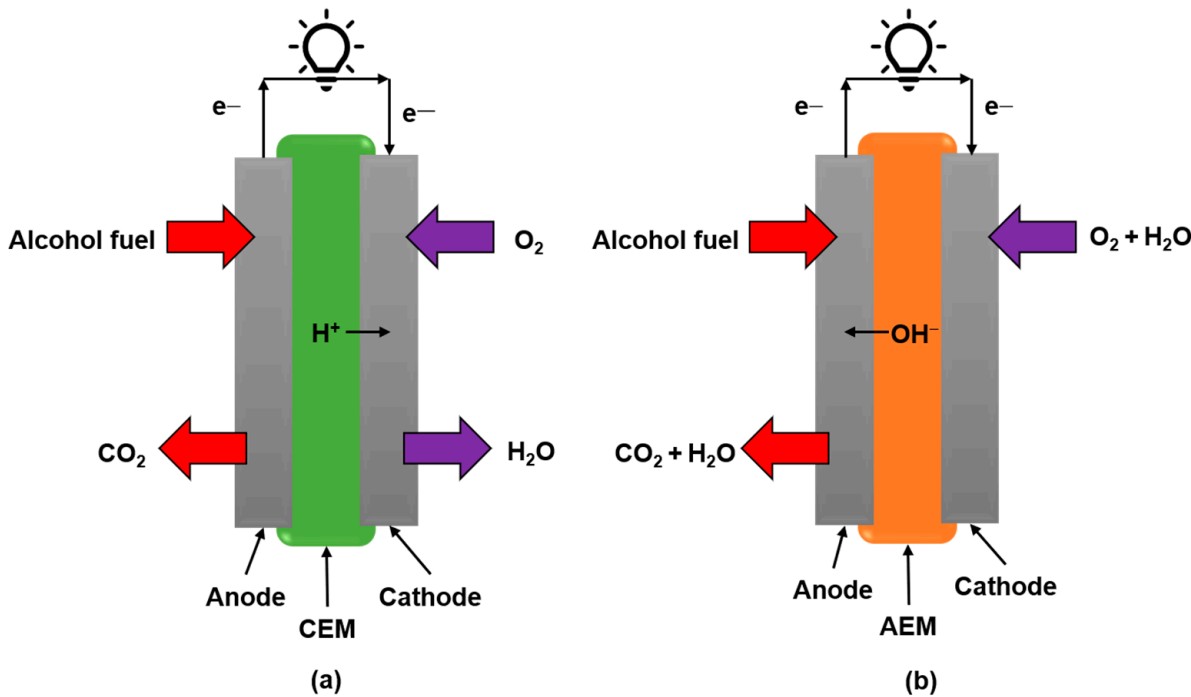

**Figure 1.** Alcohol fuel cells using (**a**) CEM and (**b**) AEM.

## 2. An Overview of Ascorbic Acid

AA (vitamin C) is a water-soluble antioxidant that is present in foods and biological systems [8,9]. The molecular formula for AA is $C_6H_8O_6$ and its molar mass is 176.14 g per mol [10]. Vitamin C or AA is the L-enantiomer of ascorbate; the D-enantiomer is the reverse and has no physiological importance. The same chemical structure exists in each of its forms, which are exact replicas of one another. Figure 2a illustrates the chemical structure of AA. One of the most pervasive vitamins ever identified is AA. It must be consumed from outside fruits and vegetables because it cannot be produced in the human body. AA is largely present in citrus fruits and vegetables [11]. AA is widely used in the chemical sector and cosmetic and medical applications due to its antioxidant properties. The mechanism and kinetics of AA oxidation have been extensively studied during the past few decades [8,12,13]. Two $e^-$ and two $H^+$ are released during AA oxidation to create dehydroascorbic acid (DHA). As seen in Figure 2b, AA and DHA form an irreversible redox pair [11]. It has been established that the production of DHA requires two $e^-$ and two $H^+$ during the oxidation of AA [14,15].

**Figure 2.** (**a**) Chemical structure of ascorbic acid and (**b**) ascorbic acid oxidation to produce dehydroascorbic acid.

## 3. Catalysts for Ascorbic Acid Electrooxidation

Research output has increased in several application fields as a result of the development of nanomaterial technology. The incorporation of nanomaterials into electrocatalyst technology improves catalysis performance by increasing sensitivity, selectivity, stability, and repeatability. The discovery of novel catalysts enables comprehensive catalyst downsizing, improve stability, and enhanced specificity. The noble metals are more stable at room temperature and have better catalytic activity and biocompatibility when compared to other nanoscale systems. Numerous applications, such as electrocatalysis, environmental monitoring, and biosensing, have previously demonstrated the value of noble metals. Numerous nanomaterials (noble metal nanoparticles, carbon nanocomposites, transition metals oxides and hydroxides, and conducting polymers) have been used in the electrocatalysis of AA [16]. The electrocatalysis of AA has been carried out using noble metals (gold, ruthenium, etc.) through high selectivity and low overpotential [17–21]. Noble-metal-modified electrochemical surfaces have a greater electrochemically active surface area, which results in a faster electron transfer rate, faster current responsiveness, and a lower overpotential required for electrochemical reactions. Pt has drawn more interest because it has better mass transport characteristics and allows for rapid electron transmission. Pt has been applied for the electrocatalysis of AA [22–25]. There are several sites for AA oxidation, hydrogen adsorption, reverse adsorption, and dissociative side-chain adsorption in the nanostructured Pt. The exorbitant price of the Pt catalysts might prevent commercialization. The use of Au nanoparticles in electrocatalysis is frequently widespread because of its electron-rich core, biocompatibility, constrained size, and several surface functions [26]. Many studies have discussed the use of Au-modified electrodes for AA electrooxidation [11,13,17,27,28]. However, the performance of Au nanoparticles for AA electrooxidation is insufficient for practical applications. Pd has high catalytic activity for electrochemical reactions. Several Pd-based catalysts for AA electrooxidation have been reported [9,11,29,30]. Ag has been employed for AA electrooxidation because of its unique electrical and chemical characteristics [8,31–36]. Non-noble catalyst Cu has been employed for AA oxidation [19,37]. Nonmetallic catalysts, such as carbon catalysts, showed prominent results for AA electrooxidation [38–44].

## 4. Ascorbic Acid as Fuel

AA, generally referred to as vitamin C, has been a subject of interest in a variety of scientific disciplines. In particular, AA has been used for DLFCs throughout the past few decades. It is a tiny organic molecule that is plentiful in living things and is, therefore, a sustainable fuel source. Additionally, AA can be produced commercially by fermenting D-glucose or from biomass waste [45]. AA is oxidized effectively on a variety of carbon black materials, making it a desirable fuel choice for DLFCs [41,44]. Thus, the potential of carbon electrodes for the AA fuel cell is worth investigating. As mentioned above, the primary end product of the AA oxidation reaction has been identified as DHA, regardless of catalyst [46,47]. The creation of DHA from AA fuel and solid carbon catalyst benefits

the environment and creates a complete zero-emissions energy-harvesting system. These advantages have significant effects on AA's capacity to be used in implantable medical devices. Additionally, it has been demonstrated that DHA can be enzymatically converted to AA via a variety of pathways, suggesting effective recovery of fuel from waste [48].

### 4.1. Acidic Fuel Cell

Combining thin-film and microelectromechanical systems (MEMS), AA could be used in portable or small acidic fuel cells. According to Mogi et al., direct AA fuel cells (DAAFCs) with microchannels made of polyimide (PI) substrates have been produced [49]. On the PI substrate, aluminum electrodes were created using photolithography and screen printing. Porous carbon was deposited on the Al electrode using screen printing to improve surface area to produce 1.83 $\mu W/cm^2$ power density. These devices showed excellent versatility and endurance while serving as examples of micropower sources for in vivo sensing [49]. Falk et al. demonstrated an AA-based biofuel cell without a membrane for glucose sensing [50]. The cell produced electricity using AA and oxygen from the human lachrymal fluid. Tetrathiafulvalene (TTF) and tetracyanoquinodimethane (TCNQ) were added to nanostructured microelectrodes to act as the anode catalysts for AA oxidation and a redox enzyme called bilirubin oxidase (BOx) as the cathode catalyst. The device generated a peak power density of 0.72 $\mu W/cm^2$ and a current density of 0.55 $\mu A/cm^2$. This technology demonstrated a secure, dependable source of power for noninvasive glucose testing [50]. Wu et al. demonstrated a small silicon wafer fuel cell where a thin polymer electrolyte separator was used between AA and air reactants. The anodic catalyst for the electrooxidation of AA was a thin Pt layer (0.43 $mg/cm^2$) deposited by sputtering on a silicon surface. Using DuPont's Nafion®117, the maximum output power density was 1.95 $mW/cm^2$ at a current density of 10 $mA/cm^2$ [51].

In contrast to fuels like ethanol, methanol, and/or polyalcohol fuels, investigations have shown that the electrooxidation of AA fuel may be carried out on a cheap, non-noble metal catalyst. Using a CEM and several anode catalysts, including carbon black (Vulcan XC72) [44,45], Pt [45,47], Ru [45,47], Pd [44–47], Ir [45,47], Rh [45,47], and PtRu [45,47], Fujiwara et al. showed an operational DAAFC. Pd black metal catalyst displayed excellent power production of 7 $mW/cm^2$ among various precious metal catalysts. The highest power density was generated using Vulcan XC72 (16 $mW/cm^2$), as shown in Figure 3. Variations in fuel cell performance on each anodic catalyst are caused by differences in the electrochemically active surface area of electrodes. According to Mondal et al., a polyaniline (PANI) catalyst might be employed for DAAFC without a platinum-group catalyst. The DAAFC generated a peak power density of 4.3 $mW/cm^2$ and a current density of 15 $mA/cm^2$ at 70 °C. Kaneto et al. reported single-walled carbon nanotubes (SWCNTs) and poly(3,4-ethylene dioxythiophene) polystyrene sulfonate (PEDOT*PSS) (* denotes charge transfer complex) composite anode catalyst for DAAFC. Coating SWCNT with PEDOT*PSS enhanced peak power density to 11.3 $mW/cm^2$ higher than SWCNT (4.9 $mW/cm^2$) alone [52]. The OCV was 0.55 V. Uhm et al. found that electrochemically oxidized carbon electrodes had a considerably higher electrocatalytic performance for AA oxidation than unmodified carbon. Furthermore, the carbon paper-based electrode structure is much more effective in generating a power density of 18 $mW/cm^2$ at 60 °C without the need for an extra powder-based valuable catalyst layer [38]. To produce an anodic electrocatalyst with better hydrophilicity for AA oxidation, Qiu et al. demonstrated that BP 2000 carbon black treated with 4 M nitric acid yielded a peak power density of 31 $mW/cm^2$ at 80 °C [43]. A hydrophilic carbon produced 1.72 times the power density of the previously reported carbon catalyst. In contrast to platinum group metals, Choun et al. found that carbon surfaces with high oxygen-to-carbon ratios (O/C) demonstrated exceptional electrocatalytic activity toward AA oxidation. They discovered that acid-treated carbon on the Toray-060 catalyst produced a maximum power output of about 53 $\mu W/cm^2$ at 36.5 °C (implantable medical technology) [39].

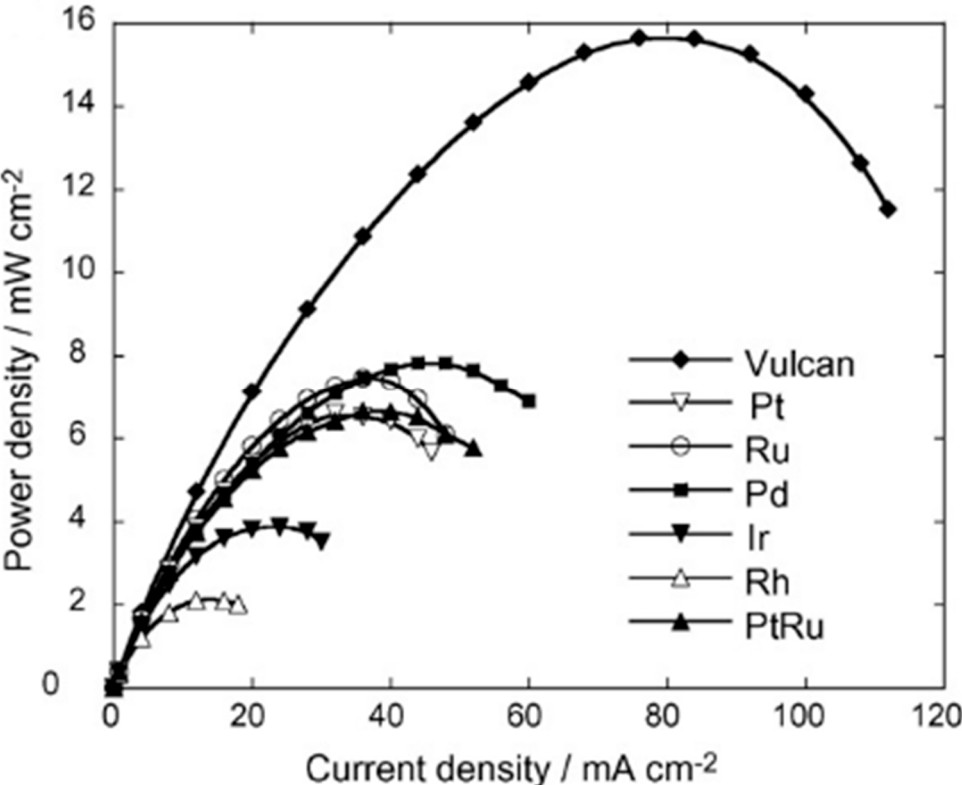

**Figure 3.** The relationship between power density and current density for DAAFCs using Vulcan XC72, Pt, Ru, Pd, Ir, Rh, and PtRu anode catalysts and Pt-PTFE black as cathode catalysts at room temperature using Nafion 117 membrane. Reproduced with permission [45]. Copyright 2007 Elsevier B.V.

### 4.2. Alkaline Fuel Cell

The AEM used in alkaline DLFCs has benefits over the CEM (e.g., Nafion) [53]. Alkaline DLFCs produce water at the anode, minimizing the need for water management, and fuel cell performance does not affect the cathode. They have faster reaction kinetics, lower activation overpotential requirement, and a lower corrosive environment for metal catalysts of the fuel cell. Lastly, compared to hydrogen, alkaline DLFCs are easier to store, handle, and transfer [54]. According to Muneeb et al., ascorbate is more effectively oxidized in alkaline media when used with an alkaline DAAFC. The initial direct alkaline DAAFC was operated by Muneeb et al. using an anion exchange membrane. The DAAFC showed a peak power density of 27 mW/cm$^2$ at room temperature using Pd black and Pt black catalysts, higher than the acidic AA fuel cell previously described. At 60 °C, this DAAFC generated a maximum power density of 73 mW/cm$^2$ and a current density of 497 mA/cm$^2$, as shown in Figure 4 [55]. Additionally, Muneeb et al. discovered that the use of a Pd$_{28}$Cu$_{72}$/C catalyst further improves the performance of alkaline DAAFC, followed by a maximum power density of 89 mW/cm$^2$. This is because of the bimetallic advantage of the electrical and bifunctional effects [37]. This is very important because, by substituting roughly 3/4 of the Pd with Cu, higher performance and lower fuel cell material costs were also attained. The current work examines the idea of using a carbon-supported Cu catalyst in place of precious metal catalysts at the anode entirely.

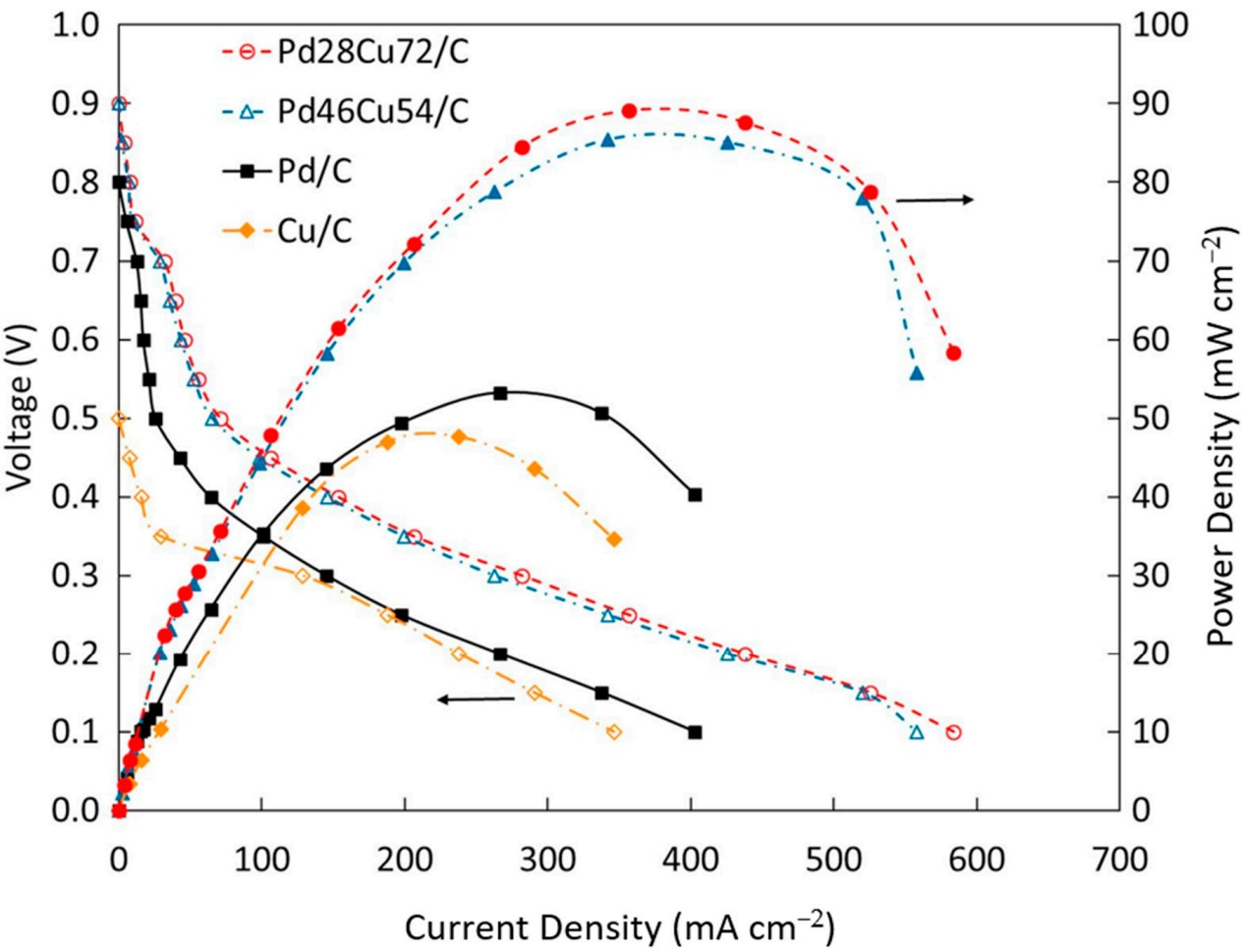

**Figure 4.** Voltage–current plots of alkaline DAAFC at 60 °C with 1 M ascorbate + 1 M KOH (anode) and oxygen (cathode). Reproduced with permission [37]. Copyright 2018 Elsevier Ltd.

### 4.3. Split-pH (Alkaline–Acid) Fuel Cells

The split-pH DLFC was proposed by An and Zhao et al. They investigated the DEFC with an alkaline ethanol fuel in anode and an acidic oxidant flow ($H_2O_2$ and $H_2SO_4$) on the cathode side with a CEM (Nafion) as an electrolyte [56]. A proton usually migrates from the anode to the cathode in DLFCs with a CEM. However, protons in CEM are replaced by Na+ (or K+) ions in split-pH fuel cells, allowing CEMs to be employed in an alkaline condition (NaOH), as shown in Figure 5a. It has been demonstrated that the oxidation of several fuels, such as methanol, ethanol, formate, and ascorbate, happens most effectively in alkaline media, but AEMs have many drawbacks when compared to CEMs. AEMs are less thermally stable and have around half or 1/3 lower ionic conductivity than CEMs [57]. As a result, the split-pH fuel cell integrates the conductivity advantages of CEM and the kinetic benefits of alkaline fuel conditions into a single system. Additionally, the use of $H_2O_2$ (oxidant) in DLFC extended for use in space and underwater travel [58]. In alkaline conditions, $H_2O_2$ also has a larger reduction potential than $O_2$ (+0.48 V higher), which rises to +0.91 V by the acidic oxidant. As a result, split-pH fuel cells have a thermodynamic advantage of greatly enhanced theoretical cell potential. The fuel showed a significantly large oxidation potential at pH 14 and a high reduction potential of $H_2O_2$ at pH 0 [59]. Li et al. tuned a formate–peroxide DLFC to achieve a peak power density of 591 mW/cm$^2$ (highest until now) having 1.69 V OCV at 60 °C by Pd/C anode catalyst and Pt/C cathode catalyst [60]. With a high power density of 33 mW/cm$^2$ at 60 °C, they also demonstrated that an AEM-based formate–oxygen DLFC might run without extra hydroxide [61]. Formate–oxygen

DLFC and formate–peroxide DLFC have a significant power density difference, which highlights the excellent efficiency of split-pH fuels.

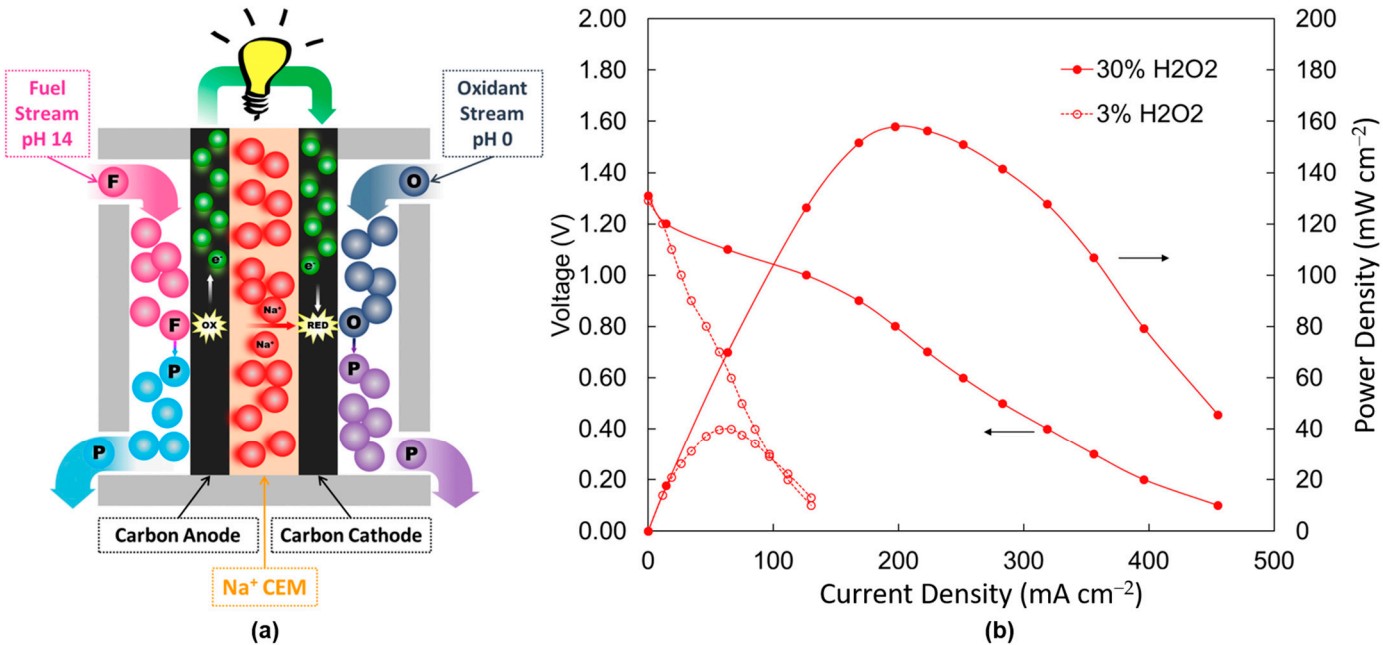

**Figure 5.** (**a**) Working principle of the split-pH DAAFC, (**b**) VI plots of the split-pH DAAFC with anode fuel 1 M ascorbate + 1 M NaOH (pH 14) and cathode oxidant is 3 or 30% $H_2O_2$ + 1 M $H_2SO_4$ (pH 0). Reproduced with permission [41]. Copyright 2018 Elsevier B.V.

Muneeb et al. showed that a split-pH DAAFC (ascorbate–peroxide) operated by carbon anode and cathode generated a peak power density of 40 mW/cm$^2$ at 60 °C using 1 M $H_2SO_4$ and 3% $H_2O_2$ [41]. This was carried out to reduce the price of fuel cell components. In addition, using 30% $H_2O_2$, this fuel cell produces the highest power density of 158 mW/cm$^2$ shown in Figure 5b. They also showed that, using the same catalyst, an alkaline ascorbate fuel cell using an AEM generated 4.7 mW/cm$^2$ peak power density with $O_2$ and 16 mW/cm$^2$ with 30% $H_2O_2$ at 60 °C. This demonstrated the performance advantages of a split-pH DAAFC over an alkaline DAAFC where the carbon electrode could be used on both sides of the fuel cell. The first nonprecious metal split-pH ascorbate–peroxide fuel cell was demonstrated by Keramati et al. They used Cu/C (anode), and carbon (cathode) as catalysts and the oxidant was $H_2O_2$ in $H_2SO_4$. It attained an OCV of 1.05 V and a maximum power density of 51 mW/cm$^2$ [59]. An alkaline environment may speed up ascorbate ($A^{2-}$) oxidation to DHA by reducing nonredox steps compared to AA oxidation at normal pH. They propose the fuel cell reaction as follows [59]:

Anode: $A^{2-}$ = DHA + 2e$^-$; $E_{anode}$ = −0.42 V
Cathode: $H_2O_2$ + 2H$^+$ +2e$^-$ = 2$H_2O$; $E_{cathode}$= +1.78 V
Overall: $A^{2-}$ + $H_2O_2$ = DHA + 2$H_2O$; $E_{overall}$= +2.18 V

The reported AA-based fuel cells are all listed in Table 1. The order is based on the highest performance.

**Table 1.** Summarized data of AA-based fuel cell.

| Anode (Catalyst Loading) | Anode Fuel | Flow (mL/min) | Cathode (Catalyst Loading) | Cathode Fuel | Flow (mL/min) | Membrane | Temp. (°C) | Peak Power Density (mW/cm$^2$) | Current Density (mA/cm$^2$) | Ref. |
|---|---|---|---|---|---|---|---|---|---|---|
| Acid-treated Vulcan XC-72 (2 mg/cm$^2$) | 1 M AA and 1 M NaOH | 2 | Acid-treated Vulcan XC-72 (2 mg/cm$^2$) | 30% H$_2$O$_2$ and 1 M H$_2$SO$_4$ | 5 | NaOH treated Nafion 115 CEM | 60 | 158 | ~475 | [41] |
| PdCu/C (2 mg/cm$^2$) | 1 M AA and 1 M KOH | 1 | Pt black (4 mg/cm$^2$) | O$_2$ | 100 | Tokuyama A201 AEM | 60 | 89 | ~560 | [37] |
| Pd catalyst (4 mg/cm$^2$) | 1 M AA and 1 M KOH | 5 | Pt black (2 mg/cm$^2$) | O$_2$ | 100 | Tokuyama A201 AEM | 60 | 73 | 497 | [55] |
| Cu/C (5 mg/cm$^2$) | 1 M AA and 2 M NaOH | 1 | Carbon (5 mg/cm$^2$) | 3% H$_2$O$_2$ and 1 M H$_2$SO$_4$ | 2 | NaOH treated Nafion 115 CEM | 60 | 51 | ~240 | [59] |
| Acid treated carbon (1 mg/cm$^2$) | 1 mM AA + 0.5 M H$_2$SO$_4$ | 15 | Pt/C (1 mg/cm$^2$) | O$_2$ | 200 | Nafion 211 | 80 | 31 | ~180 | [43] |
| Oxidized Carbon (SGL 35AA) (no metal loading) | 1 M AA | 5 | Pt on Carbon (SGL 35BC GDL) (3 mg/cm$^2$) | O$_2$ | 200 | Nafion 115 CEM | 60 | 18 | ~90 | [38] |
| Vulcan XC72 on PTFE sheet (3 mg/cm$^2$) | 0.5 M AA | 4 | Pt black on PTFE sheet (3 mg/cm$^2$) | O$_2$ | 100 | Nafion 117 CEM | 25 | 16 | ~115 | [45] |
| Vulcan XC72 (0.3 mg/cm$^2$) | 0. 5 M AA + 0.5 M H$_2$SO$_4$ | 4 | Pt–PTFE black (3 mg/cm$^2$) | O$_2$ | 100 | Nafion-117 CEM | 25 | 15 | ~85 | [44] |
| aSWCNT@PEDOT*PSS (1 mg/cm$^2$) | 0.5 M AA | 2 | Pt black (3 mg/cm$^2$) | O$_2$ | 100 | Nafion 117 CEM | 25 | 11.3 | ~60 | [52] |
| Poly aniline on TGPH 090 (35 mg/cm$^2$) | 1 M AA in 0.5 M H$_2$SO$_4$ | - | Pt on TGPH 090 (no information) | O$_2$ | - | Nafion 117 CEM | 70 | 4.3 | 15 | [62] |
| Acid treated Carbon on Toray-060 (1 mg/cm$^2$) | 0.5 M AA | 4 | Pt/C on (SGL, 10BC) (1 mg/cm$^2$) | O$_2$ | 100 | Nafion 115 CEM | 36.5 | ~0.053 | ~0.66 | [39] |

## 5. Challenges and Future Scope

AA is a natural, plentiful, and eco-friendly fuel source. However, the performance of fuel cells is still far behind for commercial use. AA functioned as a proton conductor for acidic DAAFC, much like the proton exchange ionomer Nafion. Increasing the thickness and/or mass of the Nafion ionomer reduced the AA oxidation rate by preventing AA's mass transfer towards the electrode surface [44,46]. AA is a water-based fuel. Therefore, increasing the hydrophilicity of the membrane electrode assembly (MEA) is crucial for improving fuel cell performance. The water management process might be enhanced by alkaline DAAFC. Thus, fuel cell performance was improved. However, the AEM deterioration could be troublesome for long-time operations. The benefits of the alkaline anode side and acidic cathode side in split-pH DAAFC showed the highest performance, similar to the alcohol-based DLFC. Although split-pH DAAFC significantly improved the catholyte, the higher corrosivity brought by utilizing a more concentrated $H_2O_2$ solution may cause fuel cell components to prematurely degrade. Therefore, to produce a higher amount of power density for the practical use of environmentally friendly DAAFC, the best conditions paired with a suitable anode catalyst, such as Pd- and/or Cu-based carbon catalysts, need to be further explored.

## 6. Conclusions

This review paper covered the operation of the DAAFC and the possibility that AA might be used as a renewable alternative fuel. Numerous studies demonstrated that the DAAFC produced clean energy without emitting any hazardous materials into the environment. As an alternate power source to alcohol-based DLFC, split-pH DAAFC offers similar performance. Thus, this review may provide future academics with a chance to draw attention to DAAFC improvement and build a sustainable society for the next generation.

**Funding:** This research received no external funding.

**Institutional Review Board Statement:** Not applicable.

**Informed Consent Statement:** Not applicable.

**Data Availability Statement:** Not applicable.

**Conflicts of Interest:** The author declares no conflict of interest.

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
