# Peer review of "Environment-Friendly Ascorbic Acid Fuel Cell"

_2673-3293, doi:10.3390/electrochem4010003_

Round 1

Reviewer 1 Report

In this review, author summarized the status of a new type of fuel cell that uses ascorbic acid as fuel. This review provides a refreshing perspective and comprehensive knowledge. But the author still needs to pay attention to some problems to improve the quality of the article. Please see the comments below.

Page 1 line 36, How author can conclude formic acid is less hazardous than methanol? Any reference for it?

Page 3 line 77, What’s ‘oxidation wave’?

Page 3 line 102, Pure Au doesn’t have high surface-to-volume ratio. Au nanoparticles may have this property.

Page 4 line 129, ‘converted from AA to DHA’ to ‘recover fuel’?

Page 4 line 139, 1.83 μW/cm2 power density really surprised me. What’s the total power that such kind of device can output and what’s the power requirement to drive practical applications?

Page 4 line 150, How thin the polymer electrolyte is it? What kind of polymer is it and what’s the IEC of it? What’s the Pt loading of it? Did any backpressure applied to air?

Page 5 line 164, PANI performed as what role so it doesn’t need Pt?

Page 5 line 169, What’s normal theoretical open circuit voltage for DAAFC? What’s the specific surface area of the SWCNT?

Page 5 line 183, Did author mentioned why they chose 36.5C this special temperature?

Page 6 line 185, Don’t need to write ‘Plots showing’

Page 7 line 223, Not really true. Recently many researchers developed novel kinds of AEMs and most of them have lower conductivity than PEM but just around half or 1/3 of PEM, not ‘one-order of magnitude’. Please refer "Electrospun composite proton-exchange and anion-exchange membranes for fuel cells." Energies 14.20 (2021): 6709.

Page 7 line 233, What’s the running temperature?

Page 8 Table 1, Line number 257, 258, 259 located at the top left corner of Table. The abbreviation for temperature should be ‘temp’. I think it should be called 'peak power density' if there is only one value.

Page 8 Table 1, It makes no sense to just write the type of cathode and anode without loading. Especially if the authors also mentioned the power density. Even with the same kind of catalyst, different loadings can lead to large power density differences.

Page 8 Table 1, For the data from Ref [39], how the 0.66mA/cm2 current density to get 53mW/cm2 power density?

Author Response

Thank you for your suggestions and comments. The author updated the manuscript according to all your comments. Please check the response for 'reviewer-1 comments' file for the point-by-point response to your queries and suggestions. 

Reviewer 2 Report

-Line 63: change "molecular name" by "molecular formula"

-Line 131: I think that reference 48 does not correspond to the expression described

-Lines 135, 143, 159, etc : Mogi et al. ; Falk et al. and several other references should include corresponding numbers

-Delete the space between lines 247 and 248

Author Response

Thank you for your suggestions and comments. The author updated the manuscript according to all your comments. Please check the 'response for reviewer-2 comments' file for the point-by-point response to your queries and suggestions. 

Round 2

Reviewer 1 Report

The author has modified the article as required, and my suggestion is to accept it.